# Comparison of Volatile Compounds Contributing to Flavor of Wild Lowbush (*Vaccinium augustifolium*) and Cultivated Highbush (*Vaccinium corymbosum*) Blueberry Fruit Using Gas Chromatography-Olfactometry

**DOI:** 10.3390/foods11162516

**Published:** 2022-08-20

**Authors:** Charles F. Forney, Songshan Qiu, Michael A. Jordan, Dylan McCarthy, Sherry Fillmore

**Affiliations:** 1Kentville Research and Development Centre, Agriculture and Agri-Food Canada, 32 Main Street, Kentville, NS B4N 1J5, Canada; 2Development Center of Technology for Fruit & Vegetable Storage and Processing Engineering, Guangdong University of Petrochemical Technology, No.139, Guandu 2nd Road, Maoming 525000, China; 3AGAT Laboratories, 11 Morris Drive, Dartmouth, NS B3B 1M2, Canada

**Keywords:** blueberry, aroma-active compounds, flavor, volatile compounds, 2-Dimensional gas chromatography-mass spectroscopy, gas chromatography-olfactometry (GC-O)

## Abstract

The flavor of blueberry fruit products is an important parameter determining consumer satisfaction. Wild lowbush blueberries are primarily processed into products, but their flavor chemistry has not been characterized. The objective of this study was to characterize the aroma chemistry of lowbush blueberries and compare it with that of highbush. Aroma volatiles of lowbush blueberries from four Canadian provinces and five highbush blueberry cultivars were isolated using headspace solid-phase microextraction (SPME) and characterized using gas chromatography-olfactometry (GC-O) and 2-dimensional gas chromatography-time of flight-mass spectrometry (GC×GC-TOF-MS). Lowbush fruit volatiles were composed of 48% esters, 29% aldehydes and 4% monterpenoids compared to 48% aldehydes, 26% monoterpenoids and 3% esters in highbush fruit. Twenty-three aroma-active peaks were identified in lowbush compared to forty-two in highbush fruit using GC-O. The most aroma-active compounds in lowbush fruit were ethyl 2-methylbutanoate, methyl 2-methylbutanoate, methyl 3-methylbutanoate, ethyl 3-methylbutanoate and ethyl propanoate compared to geraniol, (*Z*)-3-hexen-1-ol, 1-octen-3-one, α-terpineol and linalool in highbush fruit. The aroma volatile composition was more consistent among lowbush fruit samples than the five highbush cultivars. Aroma-active GC-O peaks were described more frequently as “floral”, “fruity”, “sweet” and “blueberry” in lowbush than in highbush fruit. Results suggest wild lowbush blueberries would provide “fruitier” and “sweeter” flavors to food products than cultivated highbush fruit.

## 1. Introduction

The consumption of blueberry fruit and products has increased rapidly over recent years, in part due to their desirable flavor and health-promoting properties [1]. From 2018 to 2020, global blueberry production increased by over 30% [2]. Blueberry production is primarily comprised of three species of blueberries, wild lowbush (*Vaccinium angustifolium* Aiton), highbush (*V. corymbosum* L.) and rabbiteye (*V. virgatum* Aiton) blueberries. However, fresh and processed fruit are typically marketed without any differentiation of species. 

The wild lowbush blueberry is produced commercially in northeast North America with principal production in the Canadian provinces of Nova Scotia, Prince Edward Island, New Brunswick, and Quebec and in the state of Maine [3]. The fruit is produced in managed wild stands that are composed of large numbers of different naturally occurring clones resulting in variation in fruit quality characteristics. All fruit is harvested in one harvest, and after grading, approximately 99% of the fruit is frozen for later consumption or processing. Fruit is distributed internationally and is used in a wide variety of food products. Wild lowbush fruit comprises over 20% of processed fruit worldwide [2].

In contrast to wild fruit, cultivated blueberry fruit are produced in plantations of named cultivars that are clonally propagated. There are many different cultivars, each having different plant and fruit characteristics. Highbush blueberry cultivars are primarily *Vaccinium corymbosum* L., but through inter-specific hybridization, the genetics of many newer cultivars include other *Vaccinium* species. This is especially true for Southern highbush blueberries where hybridization with other native *Vaccinium* species was necessary to produce cultivars adapted to warmer climates [4]. Overall, the fruit of the highbush blueberry average approximately 4 times larger than that of the wild lowbush blueberry, having more pulp, less skin and lower anthocyanins, total phenolics and antioxidant capacity than wild lowbush blueberry fruit [5]. Approximately half of the highbush blueberry fruit produced in North America are marketed as fresh fruit and half are frozen for later marketing or processing [6].

The flavor of blueberry fruit products is an important quality parameter that influences consumer satisfaction and resulting demand [7]. The chemical composition that contributes to the unique blueberry flavor includes sugars, acids and volatile compounds. Sugars are responsible for sweetness, organic acids produce tartness and volatile compounds contribute to the unique flavor and aroma of the fruit [8,9,10,11]. Blueberry aroma depends on the interaction of dozens of volatile compounds synthesized by the fruit during ripening [12]. In highbush fruit, which has been more extensively studied, approximately 120 unique volatiles have been identified [9]. Studies have shown that the volatile composition of blueberry fruit is complex and dependent on many factors including species, cultivar, environment and cultural practices [9,13,14]. 

Among the many blueberry volatiles reported, there has been limited determination of those responsible for blueberry aroma and flavor [9,15]. Volatile compounds that contribute to blueberry flavor have been assessed through sensory evaluation of synthetic mixtures [16,17], correlation with fruit sensory attributes [18,19] and gas chromatograph-olfactometry (GC-O) [10,20]. Partial least-squares regression correlated volatile compounds collected from highbush and rabbiteye blueberry fruit with aroma, and compounds correlated with aroma included linalool, hexanal, eucalyptol, β-caryophyllene oxide, 2-heptanone, neral, 2-undecanone and 3-methyl-1-butanol [18,19]. In Southern highbush blueberries, GC-O analysis found (*E*)-2-hexenal and linalool to be the most aroma-active compounds among four cultivars, while the contribution of other aroma-active compounds was cultivar-dependent and included (*E*,*Z*)-2,6-nonadienal, (*Z*)-3-hexenal, 2-heptanol, β-damascenone, geraniol and eugenol [10]. No identification of aroma-active compounds in wild lowbush blueberries has been reported.

Currently, blueberries are treated generically with little differentiation between species when using blueberry fruit in various food products. However, differences in fruit properties among the different commercially produced blueberry species can influence product quality attributes. To optimize the consistent flavor quality of blueberry products, a better understanding of differences in the flavor chemistry of blueberry species is needed. Therefore, the objective of this study was to identify the volatile compounds contributing to the flavor of wild lowbush blueberry fruit and compare them to cultivated highbush blueberry fruit. Increasing our understanding of differences in the flavor chemistry of blueberry fruit could improve the utilization of these fruit for the production of more flavorful blueberry products.

## 2. Materials and Methods

### 2.1. Blueberry Samples

Wild lowbush blueberry fruit were commercially harvested for the fresh market from commercial fields in Nova Scotia (NS), Prince Edward Island (PE), New Brunswick (NB) and Quebec (QC) during the 2018 season. After cleaning and grading, 500 g samples of fresh fruit were collected from three different fields in NS on 15 August; from three different fields in PE on 31 August; from three samples taken at different times from a fresh-pack packing line in NB on 15 August; and from three different picking baskets from one field in QC on 22 August. Fruit samples were shipped overnight with ice to the Kentville Research and Development Centre (KRDC). Upon receiving, whole fruit were frozen in liquid nitrogen and stored at −80 °C until prepared for analysis. As a measure of fruit maturity, the sugar:acid ratio of the fruit was determined and averaged 14.4, 13.9, 14.0 and 15.2 for the fruit from NS, PE, NB and QC, respectively. 

Cultivated highbush blueberries were hand harvested at commercial maturity from commercial fields near Centreville, Nova Scotia, Canada. Approximately 500 g of fruit from each of three fields were obtained for five cultivars ‘Duke’, ‘Brigitta’, ‘Jersey’, ‘Liberty’ and ‘Aurora’. Two harvests were obtained from each field, providing six samples for each cultivar. The genetic composition of these cultivars was 100% *V. corymbosum*, except for ‘Duke’, which was 96% *V. corymbosum* with 4% being *V. augustifolium* [4]. Harvests occurred during the 2018 season on 13 and 22 August for ‘Duke’, 15 August and 10 September for ‘Brigitta’, 20 and 27 August for ‘Jersey’, 29 August and 4 September for ‘Liberty’ and 12 and 24 September for ‘Aurora’. The day of harvest, fruit were transported to KRDC, frozen in liquid nitrogen and stored at −80 °C until prepared for analysis. The sugar:acid ratio averaged 13.4, 11.0, 10.6, 7.5 and 6.4 for ‘Duke’, ‘Brigitta’, ‘Jersey’, ‘Liberty’ and ‘Aurora’ fruit, respectively.

### 2.2. Gas Chromatography-Olfactometry (GC-O)

Methods used for GC-O analysis were in compliance with appropriate laws and institutional guidelines and were approved by the Agriculture and Agri-Food Canada Human Research Ethics Committee (Approval 208-F-001). Aroma-active compounds were identified using an ODP3 olfactory port (Gerstel Inc., Linthicum, MD, USA) installed on a Varian 4000 GC-MS system (Varian Inc., Walnut Creek, CA, USA). Fruit samples were taken from a −80 °C freezer and held overnight at −20 °C. A 20 g (±0.05 g) sample of fruit was combined with 80 g (±0.05 g) of a saturated NaCl solution and homogenized using a Kinematica, model MB 800 Laboratory Mixer (Kinematica AG, Luzern, Switzerland) for 1 min at a setting of 6. The blended mixture was left to settle for 10 min and then a 10 g sample was transferred to a 20 mL headspace vial that was capped with a septa lid. Vials were placed on a CombiPAL auto sampler for olfactory analysis. Prior to analysis, sample vials were held at 30 °C for 5 min, after which a divinylbenzene/carboxen/polydimethylsiloxane (DVB/CAR/PDMS) solid-phase micro extraction (SPME) fiber (Sigma-Aldrich Canada Co., Oakville, ON, Canada) was introduced into the vial headspace and allowed to adsorb headspace volatiles for 30 min. Preliminary trials found that this fiber and these adsorption conditions produced the largest amount of volatiles, which were consistent across the entire chromatogram and was similar to methods used by Du and Rouseff [10]. The SPME fiber was then desorbed for 3 min at 250 °C in the injection port of the GC onto a StabilWAX column (30 m × 0.32 mm i.d. ×1.0 µm film thickness, Restek Corporation, Bellefonte, PA, USA). The flow rate of the helium carrier gas was 2.5 mL min^−1^ and the oven temperature was set at 50 °C for 0.2 min, then ramped at 5.0 °C min^−1^ to 190 °C resulting in a run time of 30 min. The column effluent was split 1:1 with half going to the mass spectrometer and the other half going to the olfactory port where it was mixed with a flow of 30 mL min^−1^ humidified air. Alkane standards (C_8_-C_20_) were run periodically and used to calculate retention index of aroma active compounds. 

A trained panel of nine sensory evaluators conducted the olfactory analysis. Each replication was evaluated by five panelists. Three panelists evaluated all fruit samples, whereas the remaining six evaluators each assessed one of the three replications of each province (wild lowbush) or cultivar (cultivated highbush). Only the first harvest of the cultivated highbush fruit was subjected to olfactory analysis. Olfactory responses were collected using a touch screen and proprietary software. For each peak that was smelled, the evaluator recorded peak start time, intensity rating and olfactory descriptors by touching virtual buttons on a touch screen. Odor intensity was rated on a scale of 1 to 5, where 1 was extremely weak, 2 was weak, 3 was moderate, 4 was strong and 5 was extremely strong. Evaluators described the odor smelled by selecting one or more of sixteen descriptors that had been predetermined by the panel based on preliminary training sessions. The sixteen descriptors were “fruity”, “blueberry”, “citrus”, “floral”, “green-grassy”, “herb-like”, “sweet”, “caramel”, “roasted-nutty”, “earthy-musty”, “rancid-cheesy”, “sulfury”, “acidic-vinegar”, “pungent-sharp”, “chemical” and “other”. In addition, evaluators could note additional descriptors by recording the peak start time and descriptor on a note pad. The intensity and frequency of each aroma-active peak was summarized by calculating the modified frequency (MF) values, which provided an overall measure of the compound’s contribution to aroma [21,22,23,24]. The MF values were calculated using the formula MF(%) = [F (%) × I (%)]^½^, where F (%) was the frequency of odor detection among evaluators as a % of all evaluators, and I (%) was the average intensity as a percentage of the maximum intensity (5). Aroma peaks with MF values < 25% were not considered significant contributors to blueberry aroma. The identification of aroma-active peaks was determined by matching RI values and aroma descriptions with that of pure standards, as well as peak identification by two-dimensional gas chromatography–time of flight–mass spectroscopy (GC×GC-TOF-MS).

### 2.3. 2Dimensional Gas Chromatography-Time of Flight-Mass Spectrometry

To aid in the identification of aroma-active compounds and determine volatile profiles for wild lowbush and cultivated highbush blueberry fruit, samples were subjected to GC×GC-TOF-MS. Headspace volatile samples were collected from homogenized fruit samples that were prepared as described above. Blanks were made by transferring 10 g of the saturated salt solution into a headspace vial. In addition, retention index standards were prepared by injecting 5 µL of 10 µg µL^−1^ C_8_-C_20_ alkanes into a 20 mL headspace vial. All prepared vials were immediately placed in the autosampler rack of a MultiPurpose Sampler (MPS) (Gerstel, Linthicum, MD, USA) for analysis on a Pegasus 4D GC×GC-TOF-MS (LECO, St. Joseph, MI, USA). Sample vials were then held at 30 °C for 5 min, after which a DVB/CAR/PDMS SPME fiber (Sigma-Aldrich Canada Co., Oakville, ON, Canada) was introduced into the vial headspace and allowed to adsorb headspace volatiles for 30 min. The SPME fiber was then desorbed for 3 min at 250 °C in the injection port of the GC, followed by 4 min of conditioning at 250 °C. Helium was used as the carrier gas, and the injector operated with a 1:10 split for 1 min following the introduction of the SPME fiber. This split ratio was chosen to maximize detection sensitivity while preventing saturation of the detector. The column flow was maintained at 1.4 mL min^−1^. The GC×GC system had a polar StabilWAX column (30 m × 0.25 mm i.d.×0.25 µm film thickness, Restek Corporation, Bellefonte, PA, USA) for the first dimension and a mid-polar RXI-5Sil column (0.6 m × 0.25 mm i.d. ×0.25 µm film thickness, Restek Corporation, Bellefonte, PA, USA) for the second dimension. The two columns were interfaced with a liquid-nitrogen-cooled dual-stage cryogenic modulator and the second column was located in an oven with the temperature program independent of the first-dimension column oven. The GC×GC operating conditions were optimized using Simply GC×GC ^TM^ (LECO, St. Joseph, MI, USA). The temperature program for the primary GC oven was set at 50 °C for 0.2 min, then ramped at 10.3 °C min^−1^ to 220 °C. The secondary oven was maintained 33 °C warmer than the primary oven. The modulation period, the hot-pulse duration and the cooling time between stages were set at 1.3, 0.39 and 0.26 s, respectively. The transfer line to the TOF-MS detector source was maintained at 250 °C. The ion source temperature was 250 °C with a filament voltage of 70 eV. The data acquisition rate was 200 spectra s^−1^ for the mass range of 35–300 amu. Mass calibration and tuning were conducted daily using perfluorotributylamine (PFTBA).

Compound identification was based on the retention index (RI) and similarity with the National Institute of Standards and Technology (NIST) Mass Spectral Virtual Library (ChemSW, Fairfield, CA, USA). Identifications were also confirmed using known standards when available. Data were processed using LECO ChromaTOF software (LECO, St. Joseph, MI, USA), and an estimate of the peak area counts of each compound was calculated using the LECO APEX data deconvolution/processing routine. 

To aid in the identification of aroma-active compounds that were present in low concentrations, additional analyses were conducted. To increase the sensitivity of detection, 20 g of fruit tissue was homogenized in 30 g of a saturated NaCl solution and 10 mL of this homogenate was transferred to a 20 mL headspace vial. Analysis of the headspace volatiles was conducted as described above, except the injector operated with a 1:5 split for 1 min following the introduction of the SPME fiber to the GC injection port.

### 2.4. Statistical Analysis

The volatile data for the wild lowbush fruit were collected from a designed experiment with random effects of three replicates from four different provinces, and the fixed effect was the differences between the provinces. The volatile data for the cultivated highbush fruit had three fields from five different cultivars for the random effects and the fixed effects were the cultivars. The volatile data were analyzed by ANOVA using the statistical software Genstat 16 (VSN International, Hemel Hempstead, England, UK). Differences were considered to be significant at *p* < 0.05. For analysis of the frequency of aroma descriptors, GC-O aroma peaks were restricted to those that were identified by two or more of the five evaluators that assessed each fruit sample. The total number of each descriptor for each sample was analyzed by ANOVA, and differences among provinces and cultivars and between wild lowbush and cultivated highbush blueberries were determined by the least significant difference test (LSD_0.05_). To further explore differences in the aroma profiles of the wild lowbush and cultivated highbush blueberry fruit, volatiles were totaled via chemical classification, and principal component analysis using correlations of Euclidian distances was conducted using Genstat 16. 

## 3. Results and Discussion

### 3.1. Aroma-Active Volatiles of Wild Lowbush Blueberries

Twenty-three aroma-active peaks that had average MF values > 25% were identified in wild lowbush blueberries using GC-O and represented twenty-five compounds (Table 1). Of these, eight were identified as esters, five alcohols, five ketones, four aldehydes, two terpenes and one unknown. Most aroma-active compounds were found in fruit from all four provinces at similar MF values. All twenty-three peaks were detected in fruit from NS and QC. Geraniol and 3-heptanone were not detected in fruit from PE and (*E*)-2-nonenal was not detected in fruit from NB. In addition, ethyl butanoate was only detected by GC-O in fruit from QC and had an MF value of 32.7% (data not shown). These differences may reflect genetic diversity in the wild blueberry fields from the different provinces [5]. Furthermore, differences in fruit maturity, environmental growing conditions in the different provinces and/or handling could impact aroma-active volatile synthesis and composition, which has been reported in rabbiteye and highbush blueberries [14,15,25]. 

Through GC-O analysis, the aroma-active peaks with the greatest MF values in wild lowbush blueberry fruit were identified as the esters ethyl 2-methylbutanoate, methyl 2-methylbutanoate, methyl 3-methylbutanoate, ethyl 3-methylbutanoate and ethyl propanoate, all having average MF values > 50% (Table 1). These compounds were described as “fruity” and “sweet” with three being described as “blueberry”. The four branched-chain esters were abundant in wild lowbush blueberry fruit and comprised over 29% of the total volatile compounds, while the straight-chain ester ethyl propanoate only accounted for 0.13% (Table 2). All of these esters were previously reported among headspace volatiles collected from whole lowbush blueberry fruit except for methyl 2-methylbutanoate [26]. 3-Hexanone, which was found in low concentrations, coeluted with ethyl 2-methylbutanoate and may have contributed to the “sweet” and “fruity” aroma of this peak. Additional esters detected in this study by GC-O and contributing “fruity” and “sweet” aromas included methyl 2-pentenoate, ethyl 2-methylpropanoate and methyl 3-methyl-2-butenoate. Lugemwa et al. [26] also reported these esters in lowbush blueberry fruit.

The C_6_ alcohols and aldehydes were abundant in wild lowbush blueberry fruit and contributed strong aromas (Table 1 and Table 2). (*Z)*-3-Hexen-1-ol contributed a “green-grassy” aroma as did hexanal and (*Z*)-3-hexenal. (*E*)-2-Hexen-1-ol was described as having a “rancid-cheesy” aroma, while (*E*)-2-hexenal was described as “floral” and “sweet”. These three C_6_ aldehydes comprised over 21% of the total volatiles in wild lowbush fruit, while the C_6_ alcohols accounted for <2%, but were greater contributors to the fruit aroma. (*E*)-2-Hexanal and (*Z*)-3-hexanol were previously reported in juice extracted from lowbush blueberry fruit [27]. However, none of these alcohols or aldehydes were found in headspace collected from whole lowbush blueberry fruit [26], suggesting that their formation occurred as a result of fruit homogenization. These C_6_ alcohols and aldehydes are known to be products of lipoxygenase (LOX) activity, which occurs as a result of homogenization and the addition of NaCl to blueberry homogenates reduces LOX activity [28,29]. However, differences between headspace volatile profiles of whole fruit and fruit homogenized with NaCl suggests that inhibition of LOX activity by NaCl is not absolute. Other alcohols that contributed to the aroma of wild lowbush fruit included 2-ethyl-1-hexanol that contributed “sweet” and “floral” notes and was the most abundant alcohol comprising 5.2% of the total volatiles. 1-Pentanol and (*E*)-2-nonenal also contributed to the aroma of wild lowbush blueberry fruit (Table 1 and Table 2).

Monoterpenoids that contributed to the aroma of wild lowbush blueberries were linalool and geraniol. They had similar average MF values of 45.5% and 43.4%, respectively, and both contributed “floral” and “sweet” aromas (Table 1). While both had similar contribution to aroma, linalool was found in much higher concentrations comprising over 2% of the total volatiles, while geraniol comprised <0.05%, suggesting it may have a lower odor threshold than linalool (Table 2). Cometto-Muñiz et al. [30] reported a lower odor threshold for geraniol (0.1 ppm) compared to linalool (1.0 ppm); however, other studies have not confirmed this difference [31].

In addition to 3-hexanone, several other ketones also contributed to wild lowbush blueberry fruit aroma that included 1-octen-3-one, 2-dodecanone, 1-penten-3-one and 3-heptanone (Table 1). 1-Octen-3-one coeluted with the ester methyl 2-pentenoate and had an MF value of 49%. This aroma-active peak was described as “earthy-musty” with fewer descriptors of “fruity”, which may reflect the contribution of methyl 2-pentenoate in this coelution. 2-Dodecanone had similar aroma strength and contributed “floral”, “fruity”, “citrus” and “herb-like” notes. 1-Penten-3-one and 3-heptanone were both described as “earthy-musty” and had less contribution to fruit aroma. None of these ketones have been previously reported in wild lowbush blueberry fruit.

### 3.2. Aroma-Active Volatiles of Cultivated Highbush Blueberries

Among the five highbush cultivars evaluated in this study, a total of forty-two peaks were identified through GC-O analysis with MF values > 25% in at least one cultivar (Table 3). ‘Duke’, ‘Brigitta’, ‘Jersey’, ‘Liberty’ and ‘Aurora’ each had twenty-six, twenty-one, fourteen, twenty-two and twenty-eight aroma-active peaks, respectively. Eighteen of the identified compounds comprising these peaks were determined to be monoterpenoids, nine aldehydes, seven ketones, five alcohols, four branched-chain esters, one straight-chain ester, two acids, one hydrocarbon and one unknown. Differences in aroma-active compounds among the five highbush cultivars were substantial. Of the forty-two peaks identified through GC-O, only seven had MF values > 25% in all five cultivars and twelve peaks were detected by GC-O in only a single cultivar (Table 3). Similar differences in aroma-active compounds among four Southern highbush blueberry cultivars were reported, where twenty-four of forty-three aroma-active compounds were found in all four cultivars [10].

Monoterpenoids were major contributors of “floral”, “sweet”, “fruity”, “citrus” and “blueberry” aromas in fruit of the five highbush blueberry cultivars assessed in this study (Table 3). Of all the volatile compounds identified in these highbush fruits, monoterpenoids comprised, on average, over 25% of the total volatile content (Table 4). Geraniol, α-terpineol and linalool contributed strong aromas to the fruit of all five cultivars (Table 3). Linalool was the most abundant monoterpenoid comprising, on average, approximately 9% of the total volatiles, while α-terpineol comprised approximately 2% of the total volatiles. Geraniol averaged only 0.24% of the total volatiles but was the strongest contributor to aroma, having the first or second highest MF value for all five cultivars. Linalool, α-terpineol and geraniol were previously reported in highbush blueberries [12,17,20,29,32,33]. In four cultivars of Southern highbush fruit, linalool, and to a lesser extent, α-terpineol, contributed to the aroma of fruit, but geraniol only contributed to the aroma of two of these cultivars [10]. The occurrence of the remaining fifteen monoterpenoids that were identified through GC-O in this study varied among the five cultivars. With the exception of 2,6-dimethyl-2,6-octadiene, no additional monoterpenoids were identified in ‘Jersey’ fruit by GC-O. Six of the identified aroma-active monoterpenoids were only found in fruit of one of the five cultivars. Other studies reported differences in monoterpenoid composition among cultivars [14,19,33,34,35,36], and it was suggested that differences in monoterpenoid content could be responsible for the distinct aroma of different blueberry cultivars as well as consumer acceptability [14,36].

Branched-chain esters also contributed “fruity” and “sweet” aroma to highbush blueberry fruit, but their contribution varied among the five cultivars. Methyl 3-methylbutanoate, ethyl 2-methylbutanoate and methyl 2-methylbutanoate were strong contributors to the aroma of ‘Jersey’ fruit and moderate contributors to ‘Brigitta’ fruit (Table 3). In ‘Aurora’ fruit, ethyl 2-methylbutanoate and ethyl 3-methylbutanoate were strong contributors to the aroma. In contrast, the aroma from these esters was not detected in ‘Liberty’ fruit. The GC-MS volatile profiles of these fruit found methyl 3-methylbutanoate to be the most abundant branched-chain ester, followed by methyl 2-methylbutanoate, and they comprised approximately 12% and 5% of the total volatiles in ‘Jersey’ and ‘Brigitta’ fruit, respectively, but <1% in the other three cultivars (Table 4). Other studies have shown that the contribution of branched-chain esters to highbush blueberry flavor is cultivar dependent. Qian et al. [33] found high levels of branched-chain esters in ‘Duke’ fruit but low concentrations in ‘Aurora’ and ‘Liberty’. In both Northern and Southern highbush fruit, GC-O analysis identified that all four methylbutanoates contributed “fruity” aroma notes, although the contribution of each varied among cultivars [10,20]. In fruit from six highbush blueberry cultivars, GC-MS analysis detected methyl 2-methylbutanoate and ethyl 3-methylbutanoate in all six cultivars and ethyl 2-methylbutanoate in five of the six cultivars in low or trace concentrations, but methyl 3-methylbutanoate was not detected [29]. Qian et al. [33] suggested that branched-chain esters were associated with highbush cultivars that had desirable flavor.

The C_6_ alcohols and aldehydes were strong contributors to the aroma of cultivated highbush blueberry and comprised nearly half of the volatile compounds (Table 3 and Table 4). (*E*)-2-Hexenal and hexanal comprised 27.1% and 10.1% of the total volatiles, respectively. However, (*Z*)-3-hexen-1-ol that comprised only 0.4% of total volatiles was the strongest contributor of aroma to fruit of all five cultivars based on MF values. (*E*)-2-Hexenal and hexanal, which were also strong aroma contributors, were not identified by GC-O in ‘Jersey’ fruit. Moreover, GC-O analysis did not detect (*E*)-2-hexen-1-ol in ‘Liberty’ fruit or (*Z*)-3-hexenal in ‘Brigitta’ fruit. In two cultivars of Northern highbush fruit, GC-O identified all six of these C_6_ aldehydes and alcohols but only (*Z*)-3-hexen-1-al and (*Z*)-3-hexen-1-ol made a strong contribution to aroma [20]. In Southern highbush blueberries, three C_6_ aldehydes contributed to the aroma of all four cultivars, but of the C_6_ alcohols only (*Z*)-3-hexenol was detected by GC-O in one cultivar [10]. Horvat and Senter [37] reported that (*E*)-2-hexenal, (*E*)-2-hexenol and (*Z*)-3-hexenol were key components in blueberry flavor. Other alcohols and aldehydes that contributed strong or moderate aromas in fruit of all five cultivars of highbush blueberries in this study included 2-ethyl-1-hexanol, which contributed “floral” and “sweet” aromas and 2,6-nonadienal, which contributed “green-grassy” and “floral” aromas. These compounds were previously identified in highbush blueberry fruit [12,20,29]. 

Ketones also contributed to the aroma of cultivated highbush blueberry fruit. The most aroma-active ketone present in all five cultivars was 1-octen-3-one, which contributed an “earthy-musty”, “mushroom-like” aroma (Table 3), but, on average, only accounted for 0.06% of total volatiles. 1-Octen-3-one was previously reported to contribute to the aroma of highbush fruit [10,20]. The ketones (*E*,*Z*)-2-undecanone, 2-nonanone, 2-heptanone and, to a lesser degree, 1-penten-3-one, contributed a variety of aromas including “floral” and “fruity” notes to the highbush blueberry fruit in this study; the former three compounds were previously reported in highbush blueberries [10,12,20,29]

### 3.3. Comparison of Aroma-Active Compounds in Wild Lowbush and Cultivated Highbush Blueberries

Many of the aroma-active compounds identified in wild lowbush blueberry fruit were also identified to contribute to the aroma of cultivated highbush fruit. Of the twenty-three aroma peaks found in wild lowbush blueberry fruit, nineteen were found in at least one of the five highbush cultivars assessed in this study. Aroma-active compounds that were unique to wild lowbush fruit in this study included the three esters ethyl propanoate, ethyl 2-methylpropanoate and methyl 3-methyl-2-butenoate; one alcohol 1-hexanol; and one ketone 3-heptanone. Conversely, of the forty-two aroma-active peaks identified in the five highbush cultivars, twenty-three were not found to contribute to the aroma of wild lowbush fruit. These peaks included sixteen monoterpenoids, five aldehydes, four ketones, two acids, one alcohol and one hydrocarbon. While there were many similarities in the aroma-active compounds in the fruit of these two species, the quantities of these compounds and their contribution to fruit aroma differed considerably. These differences would impact the flavor of blueberry products and should be considered in product formulation.

The volatile profile composition of wild lowbush blueberries was dominated by esters with branched-chain and straight-chain esters comprising 31.6% and 15.9% of the total volatiles, respectively (Figure 1, Table 2). The ester content of cultivated highbush blueberries was lower than that of wild lowbush fruit with branched-chain esters and straight-chain esters averaging 2.4% and 1.0% of total volatiles, respectively (Figure 1, Table 4). Aldehydes were the most abundant volatile group in highbush fruit comprising 47.7% of the total volatiles, while in wild lowbush fruit, they were the second most abundant group comprising 28.6% of the total volatiles. The second most abundant group of volatiles in highbush fruit was monoterpenoids comprising 25.8% of total volatiles. In lowbush fruit, monoterpenoids accounted for only 4.4% of the total volatiles. Similar concentrations of alcohols (8.8% vs. 6.7%), ketones (5.9% vs. 9.5%) and hydrocarbons (2.3% vs. 5.8%) were observed in the wild lowbush and cultivated highbush fruit, respectively. 

Volatile composition among chemical groups in wild lowbush fruit did not differ significantly among provinces except for straight chain esters (*p* = 0.016) that comprised only 4.4% of total volatiles in fruit from PE compared to 13.6%, 23.6%, and 19.3% in fruit from NS, NB and QC, respectively (Table 2). Greater differences in the distribution of volatile compounds among chemical groups were seen in cultivated highbush blueberry fruit (Table 4). Aldehyde composition varied significantly among cultivars (*p* ≤ 0.001), comprising over half of ‘Aurora’, ‘Brigitta’ and ‘Liberty’ total volatiles, but <43% of total volatiles in ‘Duke’ and ‘Jersey’ fruit. Monoterpenoid content ranged from 47.9% in ‘Duke’ to 9.6% in ‘Jersey’ (*p* < 0.001). Pico et al. [29] also reported the fruit of ‘Duke’ to have higher concentrations of total terpenes than the fruit of other highbush cultivars. In ‘Jersey’ fruit, branched-chain esters (11.8%) and ketones (14.7%) were more abundant than monoterpenoids (9.6%). The average composition of other fruit volatiles that differed significantly among the five cultivars studied included ketones, branched-chain esters, straight-chain esters and furans. Qian et al. [33] found three selections from the USDA blueberry breeding program that were considered to have “outstanding” flavor had higher branch-chain ester content than seven highbush cultivars. 

To further explore the differences in the volatile chemistry between wild lowbush and cultivated highbush blueberries, principal component analysis was conducted on the volatile chemical groups (Figure 2). Scores 1 and 2 accounted for 75% of the variability. Score 1 was driven by alcohol, acid, straight-chain ester and total volatile content and score 2 was driven by branched-chain ester, hydrocarbon and ketone content. Differences were seen between the wild lowbush and the cultivated highbush fruit, with the former found in the right of the plot and the latter in the left. Wild lowbush fruit from NB and QC were similar, while those from NS and PE differed reflecting differences in total volatiles, acids, alcohols and esters. Among the highbush cultivars, ‘Brigitta’ and ‘Jersey’ were similar, while ‘Duke’ differed and associated with monoterpenoid content. 

These differences in volatile profiles were reflected in the differences in aroma-active compounds. Wild lowbush blueberry aroma was dominated by esters, while cultivated highbush fruit was dominated by monoterpenoids. In wild lowbush blueberry fruit, the five most aroma-active compounds were esters (Table 1), while in highbush fruit, three of the five were monoterpenoids and none were esters (Table 2). Lowbush fruit had two monoterpenoids that ranked eleventh and twelfth of the most aroma-active compounds, while esters in highbush fruit ranked ninth, twelfth, fifteenth and twenty-fourth, and each ester was not found in all cultivars. Alcohols, ketones and aldehydes all contributed similarly to the aroma of fruit from both blueberry species, although specific differences were observed in compounds and concentrations.

The method of volatile collection and analysis can affect volatile profiles and aroma-activity assessment [13,24,38]. All methods have advantages and disadvantages, and no method is ideal. In our study, headspace volatiles were collected from fruit homogenized in saturated salt using a DVB/CAR/PDMS SPME fiber, and aroma activity was accessed by GC-O using a sensory panel that consisted of nine evaluators. SPME fibers have a degree of selectivity and have good sensitivity for compounds with low molecular weight and high volatility but may underestimate those with low volatility [24,38]. In contrast, solvent extraction of fruit and concentration of volatiles using solvent extract dilution analysis (SAFE) is more effective in capturing volatile compounds with low volatility, including acids and hydroxyl-containing compounds. Using a total extract and SAFE, Qian et al. [20] reported vanillin as an odorant in highbush blueberry, but did not report the presence of vanillin when using SPME analysis [33]. However, the total extracts obtained by solvent extraction are less reflective of the head space volatile composition that induces the olfactory response of the consumer, and valid olfactory rankings are only obtained after odor activity values are calculated using odor thresholds for each compound [24]. In our study, aroma activity was assessed using a direct intensity measure that integrated an intensity rating and the detection frequency of panelists. The panel helps to account for variability in aroma sensitivity among individuals. Additional studies could be conducted using different analytical method such as SAFE and dilution analysis to further assess the contribution of volatiles to the aroma of wild lowbush blueberry.

To further compare the sensory impact of the aroma-active compounds of wild lowbush and cultivated highbush blueberry fruit, the frequency of descriptors chosen by sensory panelists to describe the aroma-active peaks was analyzed by ANOVA and illustrated using a radar plot (Figure 3). There was a significant interaction (*p* < 0.001) in the frequency of descriptors chosen between blueberry species and the descriptor. Of the 16 descriptors provided to the panelists, “floral” was chosen most frequently followed by “fruity” and “sweet” in both wild lowbush and cultivated highbush fruit. The fourth and fifth most frequent descriptors for wild lowbush fruit were “blueberry” and “green-grassy”, whereas for cultivated fruit, “green-grassy”, “herb-like”, “rancid-cheesy” and “earthy-musty” were chosen more frequently than “blueberry”. The descriptors “floral”, “fruity”, “sweet” and “blueberry” were chosen significantly more times to describe aroma-active compounds in wild lowbush than in cultivated highbush blueberry fruit. These results suggest that wild lowbush blueberries were perceived to have a fruitier and more “blueberry-like” aroma than cultivated highbush fruit in this study.

In addition to differences in volatile profiles that contribute to flavor differences between lowbush and highbush fruit, differences in sugar and acid composition may also impact flavor differences. Wild lowbush blueberry fruit have higher sugar content than cultivated highbush fruit, and the predominant acid is quinic acid compared to citric acid in highbush fruit [11]. Quinic acid has a less tart taste compared to citric acid [39], which would contribute to a sweeter less tart flavor of wild lowbush fruit compared to cultivated highbush fruit. The fruit of ‘Duke’, which has 4% *V. angustifolium* in its parentage, was reported to have higher quinic acid content than the other four cultivars in this study [11]. However, the volatile composition of ‘Duke’ did not show additional similarities to wild lowbush fruit.

## 4. Conclusions

The aroma-active compound composition of wild lowbush fruit produced among four Canadian provinces was more consistent than that found among the five highbush cultivars assessed in this study. Wild lowbush blueberry fields are made up of a complex mixture of genotypes that are naturally occurring clones. Genotypic variation in aroma volatile composition among wild clones can be expected. However, the genotypic diversity among the large number of wild clones that were commercially harvested for this study resulted in a fairly consistent aroma composition regardless of the province of production. Aroma-active volatiles in wild lowbush fruit were dominated by esters that contributed “fruity” and “sweet” aromas. This was in contrast to the variation in aroma volatile composition among highbush blueberry cultivars, which are each a unique genetic clone. Aroma-active volatiles in cultivated highbush fruit were dominated by monoterpenoids that contributed “floral” aromas. Frozen blueberry fruit marketed as ingredients for food products are typically marketed as “wild blueberries” or “blueberries” (cultivated highbush) with no identification of cultivar in the later. The greater homogeneity of volatile composition in wild lowbush fruit suggests that they would impart more consistent flavor characteristics in food products than would be obtained using different cultivars of highbush blueberries. Wild lowbush blueberry fruit may also provide “fruitier” and “sweeter” flavors to a food product than would be obtained with cultivated highbush fruit.

## Figures and Tables

**Figure 1 foods-11-02516-f001:**
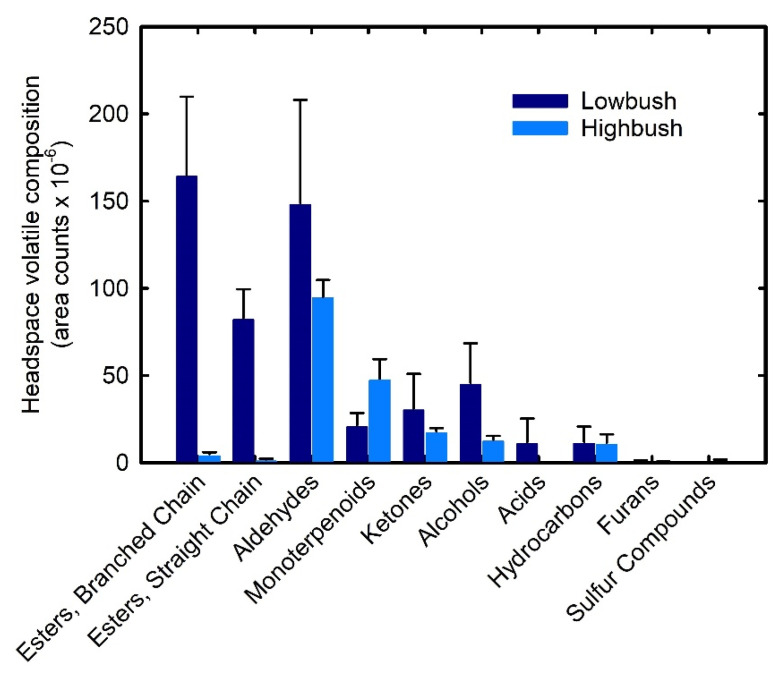
The distribution of volatile compounds in the headspace of wild lowbush and cultivated highbush blueberry fruit grouped according to chemical properties. Values represent the mean area counts collected from wild lowbush fruit samples from three commercial fields located in four provinces in eastern Canada (*n* = 12) and five cultivars of cultivated highbush fruit collected from three fields and two harvests (*n* = 30).

**Figure 2 foods-11-02516-f002:**
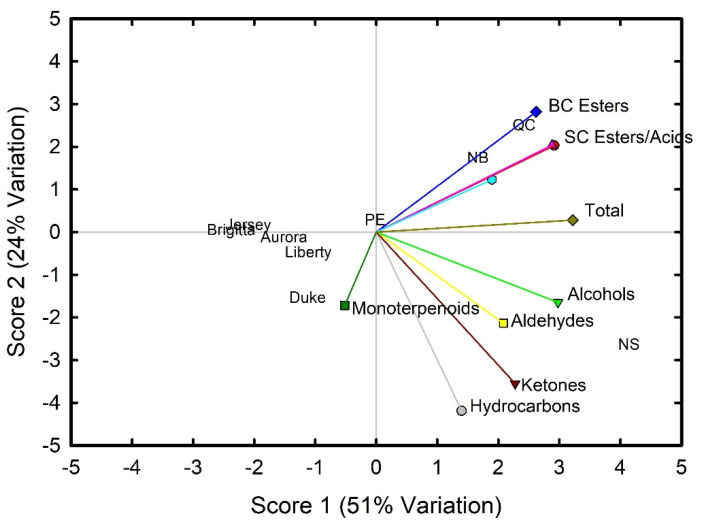
Principal component analysis of the volatile composition of wild lowbush blueberry fruit from four provinces and five cultivars of highbush blueberry fruit.

**Figure 3 foods-11-02516-f003:**
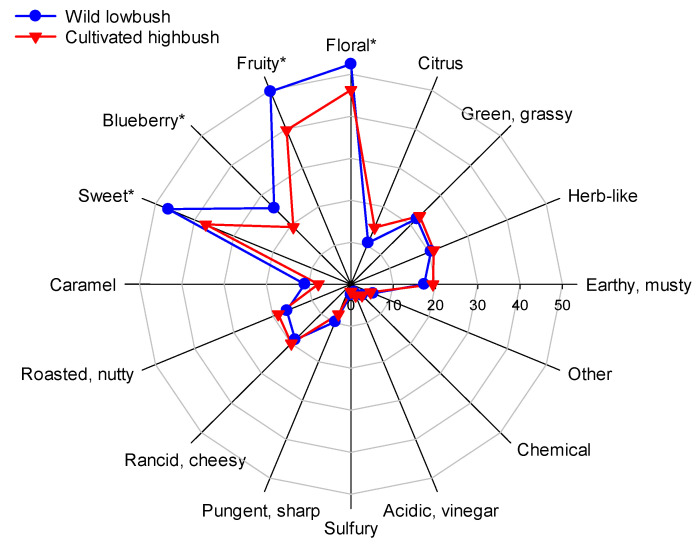
Frequency of descriptors chosen by sensory panelists to describe aroma-active compounds from wild lowbush and cultivated highbush blueberry fruit samples analyzed by gas chromatography-olfactometry (GC-O). Values represent the mean frequencies of descriptors chosen by five evaluators. Significant differences determined by LSD_0.05_ are indicated by “*” next to the descriptor. GC-O analysis was conducted on fruit collected from three wild lowbush blueberry fields located in four provinces (*n* = 12), and from five cultivars of cultivated highbush blueberries from three fields (*n* = 15).

**Table 1 foods-11-02516-t001:** Aroma-active compounds in wild lowbush blueberry fruit from the provinces of Nova Scotia (NS), Prince Edward Island (PE), New Brunswick (NB) and Quebec (QC) identified through gas chromatography-olfactometry (GC-O) and ranked by average modified frequency (MF) values ^1^.

Compound	RI	MF Value (%)					Aroma Descriptors ^2^	ID Basis ^3^
		NS	PE	NB	QC	Ave		
Ethyl 2-methylbutanoate/3-Hexanone	1052	70.6	56.6	79.2	74.0	70.1	Fruity (30) ^4^, Sweet (30), Blueberry (17), Floral (10), Citrus (4)	RI, Std, AD, MS/ RI, Std, AD, MS
Methyl 2-methylbutanoate	1012	72.1	59.6	73.0	67.1	68.0	Sweet (32), Fruity (21), Blueberry (15), Floral (12), Caramel (8)	RI, Std, AD, MS
Methyl 3-methylbutanoate	1022	64.5	60.3	49.3	66.1	60.1	Fruity (15), Sweet (14), Rancid-cheesy (12), Pungent, sharp (10), Blueberry (8)	RI, Std, AD, MS
Ethyl 3-methylbutanoate	1066	48.1	43.7	58.4	54.2	51.1	Fruity (19), Sweet (18), Blueberry (11), Floral (9), Herb-like (3), Pungent-sharp (3), Rancid-cheesy (3)	RI, Std, AD, MS
Ethyl propanoate	951	53.7	47.3	49.9	55.1	51.5	Citrus (13), Green-grassy (11), Sweet (8), Chemical (6), Fruity (6), Floral (6)	RI, Std, AD, MS
(*E*)-2-Hexen-1-ol	1401	57.5	39.5	44.7	61.1	50.7	Rancid-cheesy (19), Sulfury (13), Earthy-musty (9), Pungent-sharp (8), Roasted-nutty (8)	RI, Std, AD, MS
(*Z*)-3-Hexen-1-ol	1378	52.5	46.9	52.3	47.3	49.8	Green-grassy (18), Rancid-cheesy (10), Earthy-musty (9), Herb-like (8), Sweet (5)	RI, Std, AD, MS
1-Octen-3-one/Methyl 2-pentenoate ^5^	1303	64.5	30.1	51.6	49.9	49.0	Earthy-musty (18), Rancid-cheesy (9), Herb-like (9), Pungent-sharp (8), Fruity (5)	RI, Std, AD, MS/ RI, AD, MS
2-Dodecanone	1719	48.4	48.4	42.9	54.2	48.5	Floral (16), Fruity (10), Herb-like (10), Citrus (10), Sweet (9)	RI, Std, AD, MS
Ethyl 2-methylpropanoate	970	51.4	29.4	49.5	60.2	47.6	Sweet (23), Fruity (15), Floral (8), Citrus (6), Blueberry (4)	RI, Std, AD, MS
Linalool	1531	51.4	36.9	46.5	47.3	45.5	Floral (22), Sweet (19), Fruity (11), Herb-like (6), Blueberry (6)	RI, Std, AD, MS
Geraniol	1827	63.2	0.0	45.0	65.4	43.4	Floral (18), Sweet (10), Citrus (6), Fruity (5), Blueberry (5)	RI, Std, AD, MS
2-Ethyl-1-hexanol	1504	44.2	23.1	35.8	59.9	40.7	Sweet (11), Floral (10), Herb-like (10), Citrus (9), Green-grassy (8)	RI, Std, AD, MS
Unknown 786	786	40.0	29.7	39.0	46.2	38.7	Rancid-cheesy (18), Earthy-musty (8), Pungent-sharp (4), Blueberry (3), Floral (3), Herb-like (3)	
(*E*)-2-Hexenal	1226	20.7	51.4	21.9	48.4	35.6	Floral (10), Sweet (7), Fruity (5), Green-grassy (4), Herb-like (4), Roasted-nutty (4)	RI, Std, AD, MS
1-Penten-3-one	1031	25.3	37.9	27.1	46.2	34.1	Rancid-cheesy (9), Earthy-musty (5), Floral (3), Green-grass (3), Herb-like (3), Pungent-sharp (3)	RI, Std, AD, MS
Hexanal	1087	32.7	31.0	20.7	48.1	33.1	Green-grassy (20), Herb-like (6), Blueberry (2), Earthy-musty (2), Roasted-nutty (2), Fruity (2)	RI, Std, AD, MS
1-Pentanol	1245	32.7	30.6	21.9	47.1	33.1	Citrus (4), Roasted-nutty (3), Fruity (3), Rancid-cheesy (3), Pungent-sharp (3), Floral (3)	RI, Std, AD, MS
Methyl 3-methyl-2-butenoate	1175	38.6	24.2	25.3	37.9	31.5	Sweet (12), Fruity (11), Herb-like (5), Blueberry (3), Floral (3), Caramel (3)	RI, Std, AD, MS
1-Hexanol	1345	33.5	23.1	28.3	37.4	30.6	Fruity (11), Sweet (10), Floral (4), Blueberry (4), Carmel (2)	RI, Std, AD, MS
(Z)-3-Hexenal	1149	25.8	37.4	27.1	30.6	30.2	Green-grassy (13), Herb-like (6), Rancid-cheesy (4), Fruity (3), Floral (3)	RI, Std, AD, MS
3-Heptanone	1152	23.1	0.0	49.4	35.8	27.1	Green-grassy (14), Earthy-musty (7), Floral (4), Herb-like (3), Fruity (2), Roasted-nutty (2)	RI, Std, AD, MS
(*E*)-2-Nonenal	1541	42.2	31.0	0.0	34.6	26.9	Earthy-musty (8), Roasted-nutty (6), Rancid-cheesy (5), Sweet (4), Fruity (3)	RI, Std, AD, MS

^1^ Values for each province are the average of fifteen GC-O analyses conducted on fruit from three commercial lowbush fields by five evaluators for each sample. Compounds presented had average MF values > 25%. ^2^ The five most frequent descriptors. ^3^ RI, retention index; Std, standard; AD aroma description; MS mass spectrum (See Table 2 for RI reference comparisons and MS similarity values). ^4^ Frequency of descriptor chosen by GC-O panelists. ^5^ Tentative identification.

**Table 2 foods-11-02516-t002:** Headspace volatile composition of wild lowbush blueberry fruit from the provinces of Nova Scotia (NS), Prince Edward Island (PE), New Brunswick (NB) and Quebec (QC) determined by 2-dimensional gas chromatography–time of flight–mass spectrometry (GCxGC-TOF-MS) ^1^.

Compound	RI	RI-Ref ^2^	Similarity ^3^	Volatile Composition (Area Counts)						F Prob ^4^	%
				NS	PE	NB	QC	Grand Mean	SEM		
Acids											
Octanoic acid	2059	2051	902.2	4429	-- ^5^	6898	8099	4857	5916	ns	0.93
2-Ethylhexanoic acid	1946	1934	926.0	3587	--	2396	4468	2613	2962	ns	0.50
Heptanoic acid	1952	1953	914.5	2383	--	2442	3380	2051	2390	ns	0.39
R-4-Methylhexanoic acid	1928	1925	832.0	1096	--	981	1512	897	1042	ns	0.17
5-Methylhexanoic acid	1907	1914 ^6^	910.3	825	--	734	1204	691	812	ns	0.13
3-Methylhexanoic acid	1886	1869 ^6^	894.0	563	--	651	962	544	657	ns	0.10
Total				12,884	0.000	14,102	19,625	11,653	13,779	ns	2.24
Alcohols											
2-Ethyl-1-hexanol	1486	1491	934.0	84,188	961	12,129	10,615	26,973	24,867	ns	5.17
(*E*)-2-Hexen-1-ol	1404	1402	935.0	3385	4479	4458	9004	5332	1241	0.068	1.02
1-Hexanol	1349	1370	902.6	2966	1915	1900	7792	3643	973	0.015	0.70
Ethanol	929	929	944.0	2140	390	1335	9875	3435	2702	ns	0.66
1-Pentanol	1245	1241	895.0	1715	2169	--	3920	1951	1591	ns	0.37
3-Methyl-1-butanol	1202	1204	913.0	326	351	1716	4004	1599	651	0.022	0.31
(*Z*)-3-Hexen-1-ol	1382	1381	932.0	906	817	698	3227	1412	411	0.013	0.27
2-Hexyn-1-ol	1205	1207	885.5	1355	1477	827	267	981	721	ns	0.19
1-Heptanol	1451	1463	893.0	781	--	182	195	289	256	ns	0.06
Total				97,762	12,559	23,244	48,899	45,616	33,412	ns	8.75
Aldehydes											
(*E*)-2-Hexenal	1223	1216	940.0	27,315	101,405	43,858	38,607	52,796	15,783	0.058	10.13
Hexanal	1080	1087	911.0	53,158	40,319	27,297	24,565	36,335	14,383	ns	6.97
(*Z*)-3-Hexenal	1144	1142	870.0	10,064	34,454	30,491	19,731	23,685	13,809	ns	4.54
Heptanal	1186	1182	892.5	43,871	891	6925	4602	14,072	14,755	ns	2.70
2-Ethylhexanal	1188	1210	935.0	32,448	--	887	789	8531	11,435	ns	1.64
Pentanal	982	981	897.0	9001	2580	3061	6614	5314	2160	ns	1.02
(*E*,*E*)- 2,4-Hexadienal	1405	1414	924.9	1043	2449	1979	1472	1736	746	ns	0.33
Octanal	1291	1288	920.0	3339	524	686	487	1259	1065	ns	0.24
Nonanal	1396	1392	883.0	1281	996	1223	1123	1156	477	ns	0.22
(*E*)-2-Heptenal	1328	1319	915.0	1552	1038	505	965	1015	504	ns	0.19
3-Methylbutanal	918	912	868.0	1286	610	--	775	668	381	ns	0.13
3-Methylpentanal	1034	1032 ^7^	829.6	2212	--	408	--	655	595	ns	0.13
2-Pentenal	1133	1073 ^6^	860.6	656	382	148	499	421	396	ns	0.08
2-Methylpentanal	996	na ^8^	862.6	1135	--	243	--	345	356	ns	0.07
4-Methylhexanal	1158	na	815.0	1233	--	--	--	308	478	ns	0.06
Methacrolein	888	886.3	913.0	965	216	--	--	295	323	ns	0.06
Unknown 1165	1165	NA	845.0	1041	--	--	--	260	521	ns	0.05
Total				191,601	185,863	117,712	100,229	148,851	78,165	ns	28.55
Amines											
Dimethylamine	882	na	893.5	--	--	1497	1334	708	1083	ns	0.14
Total				--	--	1497	1334	708	1083	ns	0.14
Esters-Branched Chain											
Methyl 3-methylbutanoate	1017	1018	943.5	76,446	102,763	125,131	95,571	99,978	28,679	ns	19.18
Ethyl 3-methylbutanoate	1063	1068	948.3	13,835	13,575	28,727	83,042	34,795	15,906	0.061	6.67
Methyl 2-methylbutanoate	1012	1048	901.9	10,061	9057	24,997	13,592	14,427	4852	ns	2.77
Ethyl 2-methylbutanoate	1046	1052	939.8	3362	2575	6763	12,207	6227	3146	ns	1.19
Methyl 3-methyl-2-butenoate	1170	1170	937.0	1806	2358	3925	1022	2278	746	ns	0.44
Methyl 3-methyl-3-butenoate	1118	na	926.0	1855	1961	2593	407	1704	306	0.012	0.33
Methyl 2-methylpropanoate	921	919	889.8	600	778	3579	969	1481	366	0.004	0.28
3-Methylbutyl acetate	1120	1124	884.7	326	392	4359	99	1294	556	0.004	0.25
Methyl 3-hydroxy-3-methylbutanoate	1375	1374	884.3	523	2636	1245	271	1169	1269	ns	0.22
Ethyl 2-hydroxy-3-methylbutanoate	1426	1422	922.0	904	--	1661	514	770	403	ns	0.15
Ethyl 2-methylpropanoate	961	960	918.0	127	--	1473	1170	693	375	0.074	0.13
Total				109,844	136,094	204,453	208,864	164,814	56,604	ns	31.62
Esters-Straight Chain											
Ethyl Acetate	888	879	953.0	65,690	10,775	89,966	67,634	58,516	13,283	0.026	11.23
Methyl acetate	825	815	973.7	15,204	3424	29,942	22,563	17,783	5228	0.052	3.41
Methyl butanoate	986	983	942.0	1382	1173	2380	2994	1982	1007	ns	0.38
Ethyl butanoate	1032	1032	894.5	--	--	919	3931	1212	1564	ns	0.23
(*E*)-Hexenyl acetate	1333	1334	888.6	212	857	935	1477	870	620	ns	0.17
Ethyl propanoate	954	952	874.6	289	0	1338	1034	665	617	ns	0.13
Methyl propanoate	907	906	937.5	401	422	1561	221	651	264	0.039	0.12
Pentyl acetate	1120	1175	875.5	--	--	--	2511	628	891	ns	0.12
Methyl 2-pentenoate	1305	na	729.4	688	698	855	--	560	569	ns	0.11
Total				83,866	17,349	127,897	102,364	82,869	24,042	0.016	15.90
Furans											
2-Ethylfuran	954	949	875.0	558	511	529	411	502	421	ns	0.10
2-Ethyl-5-methyltetrahydrofuran	939	na	777.6	228	151	--	856	309	428	ns	0.06
Total				786	663	529	1267	811	849	ns	0.16
Hydrocarbons											
Toluene	1041	1042	912.5	9754	1147	--	2228	3282	2965	ns	0.63
(*Z*)-1-Ethyl-2-methylcyclopropane	1061	1062	752.5	--	--	7323	409	1933	3701	ns	0.37
Benzene	946	943	965.0	7207	--	206	0	1853	1872	0.084	0.36
Hexane	600	600	909.0	1085	1138	1729	783	1183	380	ns	0.23
m-Xylene	1144	1144	953.6	1002	2253	--	--	814	322	0.007	0.16
Ethylbenzene	1130	1141	916.4	2037	826	--	--	716	534	ns	0.14
4-Methyl-1,3-pentadiene	782	796 ^6^	946.8	227	470	876	757	583	260	ns	0.11
o-Xylene	1190	1188	934	984	975	--	--	490	511	ns	0.09
2-Octene	841	858	861.8	1217	--	--	220	359	396	ns	0.07
Indene	1495	1471	927.0	1117	--	--	--	279	282	0.073	0.05
4,4-Dimethyl-1,2-pentadiene	954	na	721.7	556	357	--	108	255	370	ns	0.05
3-Propoxy-1-propene	1007	na	865.0	938	--	--	--	235	469	ns	0.04
Total				26,124	7165	10,134	4505	11,982	12,063	ns	2.30
Ketones											
2-Heptanone	1182	1184	922.5	19,191	8673	6086	4419	9592	6738	ns	1.84
2-Nonanone	1389	1390	909.75	3152	6738	2002	7649	4885	2166	ns	0.94
3-Heptanone	1152	1151	865	9414	231	2065	616	3081	2745	ns	0.59
4-Heptanone	1124	1142	912.75	6389	--	1833	176	2100	1874	ns	0.40
5-Methyl-3-methylene-2-hexanone	1254	na	811.0	8056	--	--	--	2014	3120	ns	0.39
6-Methyl-5-hepten-2-one	1339	1340	879.7	1633	1768	1165	2348	1729	409	ns	0.33
2-Methyl-2-hepten-4-one	1214	NA	808.5	5568	--	--	--	1392	2260	ns	0.27
3-Hexanone	1049	1047	898	4281	--	226	--	1127	1325	ns	0.22
2-Butanone	904	903	843.7	4053	159	--	--	1053	1070	0.09	0.20
3-Methyl-3-buten-2-one	996	996	922.8	616	1265	--	1229	778	726	ns	0.15
1-Penten-3-one	1022	1030	881.75	664	826	186	1011	672	520	ns	0.13
2-Methylcyclopentanone	1198	1187	841.0	2125	--	--	--	531	703	ns	0.10
5-Hexen-2-one	1130	1137 ^7^	939.0	1673	--	--	--	418	564	ns	0.08
Acetone	819	817	958.5	1110	190	--	256	389	319	ns	0.07
2-Octanone	1286	1290	915.0	1499	--	--	--	375	483	ns	0.07
1-Octen-3-one	1304	1301	842.5	1446	--	--	--	361	531	ns	0.07
3-Methyl-2-butanone	932	939	826.0	1359	--	--	--	340	528	ns	0.07
3-Hexen-2-one	1218	1212	903.0	--	--	--	117	29	58	ns	0.01
Total				72,230	19,851	13,564	17,822	30,866	26,137	ns	5.92
Monoterpenoids											
Linalool	1542	1540	897.4	10,974	8268	12,813	13,949	11,501	3640	ns	2.21
(2R,5S)-2-Methyl-5-(prop-1-en-2-yl)-2-vinyltetrahydrofuran	1243	1226	887	2217	2380	4403	3021	3005	1601	ns	0.58
α-Terpineol	1698	1690	922.5	1589	924	1739	2598	1713	482	ns	0.33
(2R,5R)-2-Methyl-5-(prop-1-en-2-yl)-2-vinyltetrahydrofuran	1211	1237	778.3	1415	925	814	1660	1204	932	ns	0.23
α-Myrcene	1162	1164	894.4	956	973	1158	1378	1116	497	ns	0.21
2-(1-Hexyn-1-yl)-3-(methoxymethyl)oxirane	1325	na	735.3	504	962	1802	597	967	622	ns	0.19
Limonene	1202	1198	908.8	543	580	983	1394	875	359	ns	0.17
Eucalyptol	1209	1225	880	--	421	2375	--	699	298	0.004	0.13
β-Ocimene	1252	1248	898	1077	428	702	582	697	325	ns	0.13
Terpinolene	1286	1281	906.6	651	402	851	788	673	377	ns	0.13
p-Cymenene	1444	1439	933.1	754	478	774	648	663	276	ns	0.13
Total				20,680	16,742	28,414	26,617	23,113	9409	ns	4.43
Grand Total				615,776	396,284	541,546	531,525	521,283	255,543	ns	100.0

^1^ Values represent the mean of 3 commercial fields for each cultivar (*n* = 3). Only compounds with an average abundance >0.05% are shown. ^2^ Reference RI values are the average of 3 or more values from the National Institute of Standards and Technology (NIST) 2017 RI Database unless indicated otherwise. ^3^ MS Similarity values are the average of 3 samples unless indicated otherwise. ^4^ Significance effects among provinces based on ANOVA. ^5^ Value was below the threshold relative abundance of 0.05%. ^6^ Based on 1 value. ^7^ Based on 2 values. ^8^ na-RI not available in published databases.

**Table 3 foods-11-02516-t003:** Aroma-active compounds in cultivated highbush blueberry fruit of the cultivars ‘Duke’, ‘Brigitta’, ‘Jersey’, ‘Liberty’ and ‘Aurora’ identified through gas chromatography-olfactometry (GC-O) and ranked by average modified frequency (MF) values ^1^.

Compound	RI	MF Value (%)						Aroma Descriptors ^2^	ID Basis ^3^
		Duke	Brigitta	Jersey	Liberty	Aurora	Ave		
Geraniol	1827	68.0	69.7	75.7	74.0	66.1	70.7	Floral (38) ^4^, Citrus (18), Fruity (18), Sweet (15), Blueberry (11)	RI, Std, AD, MS
(*Z*)-3-Hexen-1-ol	1378	70.6	65.3	61.0	64.5	72.1	66.7	Green-grassy (20), Earthy-musty (18), Herb-like (16), Rancid-cheesy (14), Floral (13)	RI, Std, AD, MS
1-Octen-3-one	1303	60.2	58.5	66.3	61.1	64.5	62.1	Earthy-musty (19), Mushroom (13), Pungent-sharp (11), Herb-like (10), Rancid-cheesy (9)	RI, Std, AD, MS
α-Terpineol	1720	51.4	64.8	46.2	58.4	65.7	57.3	Floral (20), Herb-like (16), Green-grassy (14), Citrus (11), Fruity (9)	RI, Std, AD, MS
Linalool	1531	65.3	46.5	66.9	44.7	55.1	55.7	Floral (19), Fruity (18), Sweet (18),Herb-like (9), Citrus (8)	RI, Std, AD, MS
2-Ethyl-1-hexanol	1495	46.5	59.3	69.7	52.5	31.5	51.9	Floral (18), Sweet (11), Fruity (8), Citrus (8), Green-grassy (7), Herb-like (7)	RI, Std, AD, MS
2-Undecanone	1592	37.7	44.2	33.7	55.9	44.7	43.3	Green-grassy (12), Floral (11), Herb-like (9), Earthy-musty (5), Rancid-cheesy (4), Pungent-sharp (4), Blueberry (4), Roasted-nutty (4)	RI, Std, AD, MS
(*E*)-2-Hexenal/ɑ-Ocimene	1228	34.8	45.6		62.0	72.1	42.9	Sweet (16), Floral (14), Fruity (12), Citrus (5), Green-grassy (5)	RI, Std, AD, MS/ RI, Std, AD, MS
Ethyl 3-methylbutanoate	1067	37.0	50.4	26.8		60.2	42.4	Sweet (15), Blueberry (14), Fruity (14), Floral (6)	RI, Std, AD, MS
Hexanal	1088	40.0	60.8		49.3	62.0	42.4	Green-grassy (28), Floral (8), Herb-like (6), Fruity (4), Sweet (2)	RI, Std, AD, MS
(E)-2-Hexen-1-ol/2,6-Dimethyl-2,6-octadiene ^5^	1401	58.2	54.8	42.2		34.8	38.0	Rancid-cheesy (16), Roasted-nutty (9), Earthy-musty (7), Sulfury (5), Blueberry (3), Herb-like (3)	RI, Std, AD, MS/ RI, MS
Ethyl 2-methylbutanoate	1051	33.5	30.6	61.1		62.7	37.6	Sweet (16), Fruity (15), Blueberry (8), Floral (7), Citrus (2), Herb-like (2)	RI, Std, AD, MS
1-Pentanol	1244	40.4		52.5	46.2	45.4	36.9	Herb-like (7), Rancid-cheesy, (7) Earthy-musty (5), Citrus (5), Pungent-sharp (4), Floral (4), Roasted-nutty (4), Sweet (4)	RI, Std, AD, MS
Nonanal	1395	33.5	59.3		43.2	45.6	36.3	Herb-like (5), Fruity (2), Sweet (2), Floral (2), Citrus (2), Pungent-sharp (2)	RI, Std, AD, MS
Methyl 2-methylbutanoate	1014	24.2	48.1	69.7		37.7	35.9	Fruity (22), Sweet (17), Blueberry (8), Floral (6), Citrus (3), Carmel (3)	RI, Std, AD, MS
(*Z*)-3-Hexenal	1151	39.0		48.3	42.1	48.1	35.5	Green-grassy (22), Floral (9), Herb-like (7), Fruity (6), Citrus (4)	RI, Std, AD, MS
2-Nonanone	1384	56.6	17.9		48.2	39.8	32.5	Sweet (13), Fruity (12), Floral (10), Rancid-cheesy (6), Blueberry (5), Earthy-musty (5)	RI, Std, AD, MS
(2R,5S)-2-Methyl-5-(prop-1-en-2-yl)-2-vinyltetrahydrofuran ^5^/β-Phellandrene ^5^/Eucalyptol	1212	42.9	42.9		52.3	24.2	32.5	Floral (11), Sweet (7), Fruity (6), Citrus (6), Herb-like (4), Green-grassy (4), Rancid-cheesy (4)	RI, Std, MS/ RI, Std, AD, MS/ RI, Std, AD, MS
1-Penten-3-one/Ethyl butanoate	1031	21.9	24.0		39.8	67.9	30.7	Roasted-nutty (7), Herb-like (5), Earthy-musty (5), Fruity (4), Floral (3), Green-grassy (3), Rancid-cheesy (3)	RI, Std, AD, MS /RI, Std, AD, MS
(*E*)-2-Nonenal	1540		29.2		44.7	52.5	25.3	Earthy-musty (7), Rancid-cheesy (6), Roasted-nutty (5), Forest soil (4), Blueberry (3), Floral (3), Chemical (3), Herb-like (3)	RI, Std, AD, MS
2,6,6-Trimethyl-2-vinyltetrahydropyran ^5^	1111	29.2	40.0		35.8	18.9	24.8	Fruity (5), Earthy-musty (5), Herb-like (4), Rancid-cheesy (3), Roasted-nutty (3), Green-grassy (3), Citrus (3)	RI, AD, MS
2-Heptanone	1180		31.0		33.5	49.4	22.8	Sweet (6), Floral (6), Green-grassy (5), Fruity (3), Earthy-musty (3), Roasted-nutty (3)	RI, Std, AD, MS
(*E*,*E*)-2,4-Heptadienal ^5^/(E)-Linalool oxide	1468	24.5	53.7			29.5	21.5	Roasted-nutty (9), Rancid-cheesy (8), Earthy-musty (2), Blueberry (2)	RI, AD, MS/ RI, Std, AD, MS
Methyl 3-methylbutanoate	1021		43.8	56.6			20.1	Sweet (6), Fruity (5), Rancid-cheesy (4), Pungent-sharp (3), Blueberry (2), Caramel (2), Earthy-musty (2)	RI, Std, AD, MS
Unknown 787	787	55.9	34.4				18.1	Rancid-cheesy (9), Earthy-musty (7), Pungent-sharp (5), Sweet (2)	
(*E*)-2-Heptenal	1330	33.5			50.4		16.8	Floral (4), Fruity (4), Citrus (4), Sweet (3), Roasted-nutty (3)	RI, Std, AD, MS
Decanal	1491	24.5			24.5	31.6	16.1	Earthy-musty (5), Rancid-cheesy (3), Herb-like (3), Fruity (2), Floral (2), Green-grassy (2), Sweet (2)	RI, Std, AD, MS
2,2-Dimethyl propanoic acid ^5^	1572	25.3			24.5		10.0	Citrus (1), Blueberry (1), Carmel (1), Pungent-sharp (1), Roasted-nutty (1), Animal (1), cooked chicken (1), Acidic-vinegar (1), Earthy-musty (1), Herb-like (1)	RI, Std, MS
D-Carvone	1752	25.3			24.2		9.9	Floral (4), Fruity (3), Roasted-nutty (3), Herb-like (2), Green-grassy (2), Sweet (1) Acidic-vinegar (1), Earthy-musty (1), Rancid-cheesy (1)	RI, Std, AD, MS
Acetophenone	1643					45.6	9.1	Floral (2), Green-grassy (2), Blueberry (1), Fruity (1), Earthy-musty (2), Roasted-nutty (1), Chemical (1), Burnt Toast (1), Burning grass (1), Rancid-cheesy (1)	RI, Std, AD, MS
2-Dodecanone	1711					44.7	8.9	Floral (4), Roasted-nutty (3), Herb-like (3), Green-grassy (1), Pungent-sharp (1), Rancid-cheesy (1)	RI, Std, AD, MS
2-Methyl -1,4-pentadiene ^5^	1100					36.9	7.4	Roasted-nutty (2), Fruity (2), Sweet (1), Floral (1), Herb-like (1), Pungent-sharp (1), Ground coffee (1), Green-grassy (1), Chemical (1)	RI, MS
(*E*)-Isopiperitenol ^5^	1769	29.5	7.3				7.4	Sweet (3), Blueberry (2), Citrus (1), Fruity (1), Floral (1), Blackberry (1), Herb-like (1), Roasted-nutty (1)	RI, AD, MS
Geranyl acetone ^5^	1863	34.8					7.0	Fruity (3), Sweet (2), Roasted-nutty (2), Floral (1), Citrus (1), Blueberry (1), Earthy-musty (1), Horse (1)	RI, AD, MS
2,3,6-Trimethyl-1,5-heptadiene ^5^	1407					34.4	6.9	Floral (4), Sweet (2), Fruity (2), Earthy-musty (2), Citrus (1), Blueberry (1)	RI, MS
ɑ-Phellandrene 1	1170					32.7	6.5	Rancid-cheesy (3), Fruity (3), Sweet (2), Blueberry (1), Floral (1), Earthy-musty (1), Herb-like (1), Roasted-nutty (1)	RI, Std, AD, MS
Nerol	1797				29.5		5.9	Sweet (3), Floral (3), Citrus (2), Herb-like (2), Dill (1), Green-grassy (1), Acidic-vinegar (1)	RI, Std, AD, MS
Anethofuran ^5^	1509				29.2		5.8	Roasted, nutty (3), Floral (2), Rose (1), Sweet (1), Fruity (1), Earthy, musty (1), Camp Fire (1), Herb-like (1), Rancid, cheesy (1)	RI, AD, MS
(*E*)-3-Hexen-1-ol	1355				28.0		5.6	Sweet (2), Citrus (2), Foral (2), Rancid-cheesy (2), Fruity (1), Blueberry (1)	RI, Std, AD, MS
Octanal	1290					26.7	5.3	Rancid-cheesy (3), Citrus (2), Roasted-nutty (1), Sweet(1), Floral (1), Earthy-musty (1)	RI, Std, AD, MS
Limonene	1203	25.8					5.2	Earthy-musty (2), Sweet (2), Green-grassy (1), Floral (1)	RI, Std, AD, MS
5-Methylhexanoic acid	1906	25.3					5.1	Floral (2), Fruity (1), Rancid-cheesy (1), Earthy-musty (1), Sulfury (1), Chemical cleaner (1)	RI, Std, AD, MS

^1^ Values for each cultivar are the average of fifteen GC-O analyses conducted on fruit from three commercial fields by five evaluators for each sample. Compounds presented had MF values > 25% in at least one cultivar. ^2^ The five most frequent descriptors. ^3^ RI, retention index; Std, standard; AD, aroma description; MS, mass spectrum (See Table 4 for RI reference comparisons and MS similarity values). ^4^ Frequency of descriptor chosen by GC-O panelists. ^5^ Tentative identification.

**Table 4 foods-11-02516-t004:** Headspace volatile composition of cultivated highbush blueberry fruit of the cultivars ‘Duke’, ‘Brigitta’, ‘Jersey’, ‘Liberty’ and ‘Aurora’ determined by 2-dimensional gas chromatography–time of flight–mass spectrometry (GCxGC-TOF-MS) ^1^.

Compound	RI	RI-Ref ^2^	Sim ^3^	Volatile Composition (Area Counts)							F Prob ^4^	%
				Duke	Brigitta	Jersey	Liberty	Aurora	Mean	SEM		
Acids												
2-Ethylhexenoic acid	1935	1952	933	-- ^5^	227	306	--	442	195	191	ns	0.10
Total				--	227	306	--	442	195	191	ns	0.10
Alcohols												
(*E*)-2-Hexen-1-ol	1404	1402	933	4509	2895	1493	4395	4530	3564	646	0.012	1.82
(*E*)-2-Hexen-4-yn-1-ol	1221	na ^6^	856	1854	1604	685	2651	4001	2159	718	0.046	1.10
1-Hexanol	1347	1373	923	2141	1058	1531	2447	2324	1900	249	0.004	0.97
2-Ethyl-1-hexanol	1487	1492	924	1885	1827	1468	538	3080	1760	1034	ns	0.90
1-Pentanol	1245	1242	896	636	1514	1298	1414	634	1099	221	0.022	0.56
2-Hexyn-1-ol	1205	1207	886	993	960	643	1418	1174	1038	349	ns	0.53
(*Z*)-3-Hexen-1-ol	1383	1381	941	1177	149	1195	835	540	779	143	<0.001	0.40
2-Butanol	1016	1020	843	221	188	329	638	449	365	69	0.001	0.19
1-Octen-3-ol	1446	1443	868	141	288	215	242	266	230	53	ns	0.12
Cyclobutanol	1043	NA	789	131	198	172	131	284	183	51	ns	0.09
Total				13,689	10,680	9028	14,709	17,282	13,185	2155	ns	6.69
Aldehydes												
(*E*)-2-Hexenal	1222	1216	941	49,150	49,217	22,884	69,553	73,969	52,955	6302	<0.001	27.09
Hexanal	1083	1087	905	29,102	14,807	14,784	21,079	18,948	19,744	2270	0.002	10.10
(*Z*)-3-Hexenal	1148	1146 ^7^	887	31,965	1581	8590	9550	7589	11855	3119	<0.001	6.06
Pentanal	979	981	899	1800	7104	1710	5148	4975	4148	1065	0.008	2.12
(*E*,*E*)-2,4-Hexadienal	1405	1401	926	2022	469	636	1162	1068	1071	201	<0.001	0.55
(*E*)-3-Hexenal	1137	1138	866	1302	745	732	886	878	909	232	ns	0.46
Heptanal	1186	1184	909	274	1161	425	697	1338	779	231	0.018	0.40
Nonanal	1396	1389	897	990	621	298	331	233	494	276	ns	0.25
Methacrolein	881	890	844	113	610	187	333	589	366	110	0.013	0.19
4-Pentenal	1133	1129	852	88	345	143	287	755	323	59	<0.001	0.17
(*E*)-2-Heptenal	1329	1323	922	319	388	202	212	318	288	109	ns	0.15
Octanal	1291	1284	919	249	196	--	75	244	153	90	ns	0.08
(*E*)-2-Octenal	918	913	835	30	176	--	82	263	110	64	0.053	0.06
3-Methylbutanal	1435	1436	909	81	81	130	35	116	89	53	ns	0.05
Total				117,485	77,501	50,721	109,430	111,282	95,184	9472	<0.001	47.72
Esters-Branched Chain												
Methyl 3-methylbutanoate	1018	1016	933	1803	5783	12664	--	1332	4316	826	<0.001	2.21
Methyl 2-methylbutanoate	1010	1008	896	--	563	1012	--	--	315	68	<0.001	0.16
Ethyl 3-methylbutanoate	1065	1066	942	7	177	398	--	--	116	48	<0.001	0.06
Total				1809	6523	14,073	--	1332	4895	1062	<0.001	2.43
Esters-Straight Chain												
Ethyl acetate	894	893	866	331	74	3093	--	--	699	133	<0.001	0.36
Methyl acetate	858	856	857	--	1639	1288	--	409	667	140	<0.001	0.34
(*Z*)-2-Hexen-1-ol acetate	1334	1329	881	1397	337	80	665	539	604	116	<0.001	0.31
Total				1727	2051	4461	665	948	1887	320	<0.001	1.01
Furans												
2-Ethylfuran	951	950	918	364	89	220	393	481	309	93	0.06	0.16
Tetrahydrofuran	880	862	850	46	--	242	422	185	179	20	<0.001	0.09
Total				410	89	462	816	666	535	122	0.001	0.25
Hydrocarbons												
Hexane	594	600	852	10127	7083	8141	10429	8238	8804	4146	ns	4.50
Ethylcyclobutane	809	692 ^7^	886	2079	1936	1980	1994	1986	1995	712	ns	1.02
(*Z*,*Z*)-2,4-Hexadiene	738	na	933	686	51	45	260	383	285	107	0.003	0.15
Toluene	1048	1053	916	133	121	605	--	--	172	40	<0.001	0.09
(*E*,*Z*)-2,4-Hexadiene	754	na	937	23	205	62	286	157	147	73	ns	0.08
Total				13,048	9395	10,834	12,969	10,765	11,463	4888	ns	5.83
Ketones												
2-Butanone	905	903	916	21,885	5006	16,128	14,576	8948	13,309	854	<0.001	6.81
6-Methyl-5-hepten-2-one	1338	1340	884	3923	2772	657	2336	1820	2302	458	0.002	1.18
3-Ethylidene-1-methoxy-5-hexen-2-one	1325	na	731	5929	119	43	1568	--	1532	51	<0.001	0.78
Acetone	852	836	953	354	406	377	554	348	408	48	0.043	0.21
1-Penten-3-one	1022	1032	878	164	484	134	332	506	324	98	0.04	0.17
2-Heptanone	1182	1180	927	843	44	26	346	195	291	143	0.005	0.15
3-Methyl-3-buten-2-one	995	997 ^8^	935	155	368	161	61	679	285	109	0.006	0.15
1-Octen-3-one	1304	1301	842.5	35	255	--	58	219	113	71	0.066	0.06
Total				33,287	9454	17,526	19,832	12,715	17,923	1963	<0.001	9.50
Monoterpenoids												
Linalool	1542	1540	902	48,358	8419	7395	15,063	9012	17,649	1408	<0.001	9.03
Linalool acetate	1543	1548	866	31,870	--	--	34	30	6387	23	<0.001	3.27
α-Terpineol	1698	1690	933	11,103	1428	2097	3456	2993	4215	1696	0.005	2.16
(2R,5R)-2-Methyl-5-(prop-1-en-2-yl)-2-vinyltetrahydrofuran	1243	1233	905	12,803	216	105	2565	96	3157	94	<0.001	1.61
Limonene	1211	1226	892	11,375	437	543	2500	917	3154	1118	<0.001	1.61
β-Myrcene	1201	1226	885	8391	533	540	1228	707	2280	1119	<0.001	1.17
(2R,5S)-2-Methyl-5-(prop-1-en-2-yl)-2-vinyltetrahydrofuran	1162	1164	886	9476	--	--	624	--	2020	75	<0.001	1.03
α-Ocimene	1234	1238	900	6264	334	233	798	383	1602	889	<0.001	0.82
Terpinolene	1210	1204	850	6576	18	46	644	158	1489	1024	<0.001	0.76
Eucalyptol	1285	1280	898	--	4650	250	2417	38	1471	245	<0.001	0.75
1,3,8-p-Menthatriene	1234	1242	912	5427	--	--	1175	--	1320	239	<0.001	0.68
Dehydro-p-cymene	1616	1627 ^7^	888	4424	--	42	456	--	984	67	<0.001	0.50
β-Ocimene	1443	1440	915	3370	175	121	468	279	883	481	<0.001	0.45
o-Cymene	1276	1284	936	2316	--	--	258	0	515	5	<0.001	0.26
Geraniol	1844	1851	855	1614	44	78	314	336	477	247	0.001	0.24
α-Terpinene	1182	1178	858	1988	--	--	193	0	436	306	<0.001	0.22
2,6,6-Trimethyl-2-vinyltetrahydropyran	1111	1112	827	906	--	--	57	1094	411	189	<0.001	0.21
(*E*)-Geranylacetone	1849	1857	820	364	900	--	549	191	401	77	<0.001	0.21
(*E*)-Dihydrocarvone	1616	1627 ^7^	888	1594	--	--	--	--	319	0	ns	0.16
Unknown 1611	1611			1035	--	--	381	--	283	72	<0.001	0.14
2,6-Dimethyl-2,6-octadiene	1405	na	797	65	494	--	270	396	245	119	0.041	0.13
γ-Terpinene	1248	1241	864	1125	--	--	25	--	230	191	0.002	0.12
α-Phellandrene	1167	1172	890	1125	--	--	--	--	225	0	ns	0.12
P-Cymene-8-ol	1845	1839	859	705	--	--	--	--	141	0	ns	0.07
Z-Linalool oxide	1519	1513	873	91	--	--	442	--	107	13	<0.001	0.05
β-Phellandrene	1212	1212	869	39	--	--	130	25	39	19	<0.001	0.02
Total				172,402	17,647	11,448	34,047	16,655	48,273	11,115	<0.001	25.80
Sulfur compounds												
Dimethyl trisulfide	1391	1390	903	3759	--	--	--	--	752	49	<0.001	0.38
Dimethyl disulfide	1079	1069	944	2578	--	--	--	--	516	394	<0.001	0.26
Total				6337	--	--	--	--	1027	747	<0.001	0.65
Grand Total				360,259	133,672	118,910	192,484	172,128	206,647	26,301	<0.001	100

^1^ Values are means of 2 harvests from 3 commercial fields for each cultivar (*n* = 6). Only compounds with an average relative abundance >0.05% are shown. ^2^ Reference RI values are the average of 3 or more values from the NIST 2017 RI Database unless indicated otherwise. ^3^ MS Similarity values are the average of 3 samples unless indicated otherwise. ^4^ Significance of effects among cultivars based on ANOVA. ^5^ Value was below the threshold relative abundance of 0.01%. ^6^ na-RI not available in published databases. ^7^ Based on 1 value. ^8^ Based on 2 value.

## Data Availability

The data presented in this study are available on request from the corresponding author.

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
