# Peer review of "Comparison of Volatile Compounds Contributing to Flavor of Wild Lowbush (Vaccinium augustifolium) and Cultivated Highbush (Vaccinium corymbosum) Blueberry Fruit Using Gas Chromatography-Olfactometry"

_foods, 2022, doi:10.3390/foods11162516_

Round 1

Reviewer 1 Report

1. The important and key results data should be added in the abstract section.

2. The comparison of volatile compounds provided some useful information should  be introduced in the introduction section and cite more related articles.

3. The deeply analysis of volatile compounds might be provided such as PCA analysis. 

4. The effect and formation mechanism of represented volatile compounds should be discussed deeply by cited more related articles.

5. The significant Differences results should be labeled in the tables. 

6. The conclusion section should be added in the end of manuscript main text.

Reviewer 2 Report

Dear authors,

the manuscript presents an interesting topic. You used the most modern technique for determining the aromatic profile of the blueberries. However, it is not clear from the manuscript what the comparison of wild and cultivated blueberries is for. Especially, when you declare the aroma profiles of the blueberries in the previous literature. I recommend the manuscript for a major revision.

Make better use of the research aim of “commercially produced wild lowbush and cultivated highbush blueberry fruit for the production of more flavorful food products” in the title, abstract, results and discussion, and conclusion.

Complete comma before and. Uniform “headspace” vs “head space”

Were the samples taken at their best ripeness? What tests have you done to confirm maturity? - add information about it into materials and methods (part 2.1).

Complete the information about Kinematica, model MB 800 Laboratory Mixer for a setting of 6. For example input power, output power or mixer speed.

It is confusing, page 25, line 4: „Of the twenty-four aroma peaks found 4 to have MF values > 25% in wild lowbush blueberry fruit, nineteen were found in at least 5 one of the five highbush cultivars assessed in this study.“ – change the sentence. You compared lowbush fruit with highbush cultivars, the sentence does not make sense. And there are many confusing sentences there.

Figure 1 – I think there is a mistake, line 32: “two harvests” – it is incorrect with M & M, be careful and correct it.

Did you consider that the ´Duke´ variety is a 96% : 4% cultivar?

Reviewer 3 Report

The manuscript entitled „Comparison of volatile compounds contributing to aroma of wild lowbush (Vaccinium augustifolium) and cultivated highbush blueberries (Vaccinium corymbosum) using gas chromatography-olfactometry and 2-dimensional gas chromatography-time of flight mass spectrometry” describes a study aimed to compare the aroma profiles of two types of blueberries.

I really appreciated the amount of work done in this study. However, the selected methodologies are not appropriate for the selected aim.

Firstly, why did you use SPME for aroma analysis? SPME is a non-exhaustive, selective and competitive method, therefore it is not the most suitable for comparing aroma compounds. Notably, many aroma-active compounds are too polar or not volatile enough to be even absorbed into the fibre. The authors should apply the headspace analysis or the gold standard for aroma-active compounds analysis, which is SAFE. SAFE is exhaustive and gives a true representation of aroma-active compounds, not affected by displacement, saturation or competition effects. Please see e.g. the studies published by a group started by Professor Peter Schieberle in Germany to see the correct protocols for aroma-active compounds and key odorants analysis.

I do not understand this MF factor. To compare the intensity of aroma-active compounds, the flavour dilutions should be calculated. I am aware that it is a much more amount of work, but this is the only correct approach.

Please see the nomenclature for comprehensive gas chromatography: https://www.chromatographyonline.com/view/nomenclature-and-conventions-comprehensive-multidimensional-chromatography-update

The identification based on NIST and RI is not identification. It is a tentative identification.

Please remove this so-called “semi-quantification”. The “normalization” of peak areas in response to only 1 IS for a competitive method lie SPME is just incorrect. It generates biased and non-correct results. I am aware that there are hundreds of papers published with the same approach but it does not mean that it is valid. Please run a simple experiment with the same amount of IS for different standard mixtures at different concentrations and you will see that the peak area of IS will not be the same. Not for SPME. Therefore, don’t do this so-called quantification against a single IS and leave the results in peak areas for comparison, which would be more correct in this case.

The compounds in the Tables should be ordered according to the increasing RI.

Why Table 1 does not have a comparison of RI with literature but Table 2 has?

What is this “%” in Tables 2 and 4? And calculated based on what? Peak areas or these “quantifications”? If based on the second, then it is not correct.

What is the mean and SEM in Table 2 and 4? Each cultivar should have its own mean ± SD. SEM is used for population studies or if someone wants to artificially reduce the standard deviation. Please provide SD for each value, not for all studies varieties (which is super strange).

Round 2

Reviewer 2 Report

I have no additional comments.

Reviewer 3 Report

Although some changes have been applied, I am still not satisfied with the answers provided by the authors.

I still believe that flavour dillution is the correct approach for presenting the aroma intensity.

Changing name from "semi-quantification" into normalization does not solve the problem. For SPME normalization to single IS cause more bias than solution. If the data would be presented raw, as peak area, then it makes sense. 2-hexanone would not have the sam peak area in headspaces of different composiotion. Please change it back to Peak area - mean +/- SD for each compound.

Percents should be calculated in response to the total peak area of each chromatogram. Not to recalculated data in response to single IS. If such kind of normalization is supposed to be applied, then there should be much more IS, at least one for each chemical class.

As the authors wrote "The SEM is a measure of residual error in
the population experiment". And it is not a population experiment. These are 2 types of fruits. SEM should be used for human studies, not comparison of 2 berries.
